# Common Structural Pattern for Flecainide Binding in Atrial-Selective K_v_1.5 and Na_v_1.5 Channels: A Computational Approach

**DOI:** 10.3390/pharmaceutics14071356

**Published:** 2022-06-27

**Authors:** Yuliet Mazola, José C. E. Márquez Montesinos, David Ramírez, Leandro Zúñiga, Niels Decher, Ursula Ravens, Vladimir Yarov-Yarovoy, Wendy González

**Affiliations:** 1Center for Bioinformatics, Simulation and Modeling (CBSM), Universidad de Talca, Talca 3460000, Chile; yuliet.mazola@utalca.cl (Y.M.); jose.marquez@utalca.cl (J.C.E.M.M.); 2Departamento de Farmacología, Facultad de Ciencias Biológicas, Universidad de Concepción, Concepción 4030000, Chile; dramirezs@udec.cl; 3Escuela de Medicina, Centro de Investigaciones Médicas, Universidad de Talca, Talca 3460000, Chile; lzuniga@utalca.cl; 4Institute for Physiology and Pathophysiology, Vegetative Physiology, Philipps-University of Marburg, 35043 Marburg, Germany; decher@staff.uni-marburg.de; 5Institut für Experimentelle Kardiovaskuläre Medizin, Universitäts-Herzzentrum Freiburg Bad Krotzingen, 79110 Freiburg im Breisgau, Germany; ursula.ravens@uniklinik-freiburg.de; 6Department of Physiology and Membrane Biology, University of California, Davis, CA 95616, USA; yarovoy@ucdavis.edu; 7Millennium Nucleus of Ion Channels-Associated Diseases (MiNICAD), Talca 3530000, Chile

**Keywords:** atrial fibrillation, multi-target, drug promiscuity, druggable binding site, flecainide, Na_v_1.5, K_v_1.5, binding site comparison, polypharmacology

## Abstract

Atrial fibrillation (AF) is the most common cardiac arrhythmia. Its treatment includes antiarrhythmic drugs (AADs) to modulate the function of cardiac ion channels. However, AADs have been limited by proarrhythmic effects, non-cardiovascular toxicities as well as often modest antiarrhythmic efficacy. Theoretical models showed that a combined blockade of Na_v_1.5 (and its current, *I_Na_*) and K_v_1.5 (and its current, *I_Kur_*) ion channels yield a synergistic anti-arrhythmic effect without alterations in ventricles. We focused on K_v_1.5 and Na_v_1.5 to search for structural similarities in their binding site (BS) for flecainide (a common blocker and widely prescribed AAD) as a first step for prospective rational multi-target directed ligand (MTDL) design strategies. We present a computational workflow for a flecainide BS comparison in a flecainide-K_v_1.5 docking model and a solved structure of the flecainide-Na_v_1.5 complex. The workflow includes docking, molecular dynamics, BS characterization and pattern matching. We identified a common structural pattern in flecainide BS for these channels. The latter belongs to the central cavity and consists of a hydrophobic patch and a polar region, involving residues from the S6 helix and P-loop. Since the rational MTDL design for AF is still incipient, our findings could advance multi-target atrial-selective strategies for AF treatment.

## 1. Introduction

Atrial fibrillation (AF) is the most common arrhythmia worldwide [1]. Its management involves drugs to modulate ion channels’ activity in cardiac cells. Most antiarrhythmic drugs (AADs) present nowadays in clinical practice possess a strong propensity for inducing ventricular arrhythmias coupled with systemic toxicity when used for long periods [2]. Additional efforts to develop novel drugs are needed [3].

Ideally, drugs against AF are expected to be selective for atrial over other cardiac functions in order to avoid ventricular proarrhythmia [4]. This selectivity is achieved by targeting ion channels mainly expressed in atria or whose biophysical properties differ in atria versus ventricles tissues [4,5,6]. Well-known atrial-selective targets include K_v_1.5, Na_v_1.5 and the constitutively active Kir3.1/3.4 channels. They confer atrial selectivity by different mechanisms [4,7]. The Na_v_1.5 channel (and its current, *I_Na_*) is present in both atria and ventricle, but its biophysical properties are different in the atria, which confer atrial-selectivity to sodium channel blockers [4,7]. On the other side, K_v_1.5 is preferentially expressed in atria over ventricles and therefore has been one of the main targets for atria-selective drug design purposes [8]. This channel carries *I_Kur_*, the ultra-rapid delayed rectifier potassium current in atria but does not contribute to repolarizing currents in ventricles. When tested in humans, the *I_Kur_* block did not exhibit ventricular proarrhythmic activity. However, its efficacy in suppressing AF has been disappointing [9,10].

In contrast to the selective *I_Kur_* blockade, multi-channel blockers have progressed further in the clinic [11]. For example, amiodarone is one of the most effective antiarrhythmic drugs. Its action depends on a multi-target effect [2]. The advantages of a multi-channel blockade for AF are exemplified not only with a single-molecule blocker such as amiodarone but also with drug combinations [12,13]. Indeed, the need to explore drug combinations as an alternative for treating or preventing AF is gaining increasing relevance [3]. Several theoretical models were developed to study the effects of various drugs and their blockade of more than one ionic current in the setting of cardiac arrhythmia [14,15,16]. In detail, the combined *I_Na_* blockade with concomitant inhibition of rapid or ultrarapid delayed-rectifier *K^+^* currents (*I_Kr_* and *I_Kur_*, respectively) enhanced anti-arrhythmic effects compared with the *I_Na_* blocker alone [14]. Importantly, although synergistic anti-arrhythmic effects emerge from combining the *I_Na_* blocker with *I_Kur_* and the *I_Kr_* blockade, only a combination with the atrial-selective *I_Kur_* block has no effect on ventricles [14]. In support of the relevance of the *I_Na_* + *I_Kur_* blockade, Ni et al. [15] proposed with mathematical models that simultaneous blockage of these two currents produces synergism in electrically remodeled atria (which is a condition of chronic AF) without alterations in ventricles. It seems that multi-target directed ligands (MTDL) with a high degree of atrial-selectivity likely represent a favorable alternative to gain effective and safe antiarrhythmic drugs for treating AF.

In view of the above, we focused on K_v_1.5 and Na_v_1.5 given their potential as targets for MTDL design. Proteins that bind similar ligands usually have a similar structure or even share a similar binding site (BS) [17]. The K_v_1.5 and Na_v_1.5 channels belong to the voltage-dependent ion channel family and share a similar architecture and functional domains, including the voltage sensor domain and the ion-conducting pore domain (PD). In Na_v_1.5, a single chain is arranged into four different repeats or domains (DI-DIV) adopting a pseudo-tetrameric fold. Similarly, four domains are also present in K_v_1.5, but they are divided in four identical chains or subunits. To our knowledge, the comparison of the drug BS in ion channels is still limited; the first evidence of a common structural pattern in the Na_v_1.5 and TASK-1 drug BS was recently reported [18]. The comparison of the BS contributes to an understanding of the promiscuity nature of a ligand, the discovery of new MTDLs, drug repurposing and analysis of side effects [19].

In the present work, we compared the K_v_1.5 and Na_v_1.5 respective BS for their common blocker flecainide from an in silico perspective. Flecainide was chosen because the availability of structural and mutagenesis data about its BS in Na_v_1.5 and K_v_1.5, respectively [20,21]. In addition, flecainide is frequently used for the management of AF [22,23,24]. Flecainide primarily blocks the fast *I_Na_* current from the Na_v_1.5 channel and potassium channels including hERG (and its current, *I_Kr_*) [25,26] and, to a lesser extent, K_v_1.5 (and its current, *I_Kur_)* [27,28]. For K_v_1.5, flecainide blocks the *I_Kur_* current with a IC_50_ of 38.14 ± 1.06 µM [21]. In the case of Na_v_1.5, flecainide inhibition takes place with a low affinity (IC_50_ = 345 μM). However, the affinity dramatically increases (IC_50_ = 7.4 μM) when increasing the stimulation frequency as expected for use-dependent binding [29]. The flecainide blockade of hERG yielded a IC_50_ of 1.49 µM [25]. This inhibitory effect, on *I_Kr_* and *I_Na_*, occurs at lower concentrations, and it is likely the predominant effect during clinical use [25].

Here, we presented a computational workflow that allowed the flecainide BS comparison in K_v_1.5 and Na_v_1.5 based on the available rat Na_v_1.5 (rNa_v_1.5)-flecainide cryogenic electron microscopy (cryo-EM) structure and human K_v_1.5 (hK_v_1.5) functional studies for this drug [20,21]. This is the first effort to find a common structural pattern for flecainide binding in ion channels. We are beginning to gain a better understanding of how flecainide exerts its multi-target directed behavior in these atrial-selective ion channels.

## 2. Materials and Methods

### 2.1. Modeling Flecainide-K_v_1.5 Complex

We performed an ensemble docking pipeline to obtain the flecainide-K_v_1.5 complex (Figure 1). We started by building the structural ensembles using 300 frames from the last 30 ns of one apo K_v_1.5 molecular dynamics (MD) simulations (for one of the three replicas, the details of MD simulations are described below). The grid box (25 × 25 × 25 Å^3^) was set including residues Thr-479, Ile-502, Val-505 and Ile-508 which are likely involved in flecainide binding, as reported in mutagenesis studies [21]. Flecainide is protonated in the piperidine ring at a physiological pH (pKa 9.3, 99% charged at pH 7.4). Protonated R-flecainide was prepared with LigPrep; the ligand parameters and charges were added according to the OPLS2005 force field [30,31,32]. Flecainide was docked to each frame using Glide software [33]. The docked poses were scored using the XP (extra precision) scoring function [34]. For each frame, the 10 best scored poses were saved. The average linkage method implemented in the Maestro suite [35] was used to cluster the flecainide docked poses. The complex with the lowest XP docking score (−5.866 kcal/mol) from the most populated cluster was selected as the reference model for the flecainide-K_v_1.5 complex.

### 2.2. Setting up Ion Channel Systems

Three 100 ns MD simulations for each system (apo/holo K_v_1.5 and apo/holo Na_v_1.5) were executed using the Desmond v2019-1 [36] and OPLS2005 force field [30,31,32]. Initial structures for holo MDs correspond to the flecainide-K_v_1.5 model obtained herein and the cryo-EM structure of the rNa_v_1.5-flecainide complex (PDB code: 6UZ0) [20]. Apo MDs were computed using, as input, the K_v_1.5 homology-based model [37] and cryo-EM structure of rNa_v_1.5 in its apo form (PDB code: 6UZ3) [20]. Target structures were prepared before MD simulations, completing side chains, checking protonation states and minimizing the potential energy of the structures using the Protein Preparation Wizard from the Maestro suite [35]. Systems were embedded into a pre-equilibrated POPC (1-palmitoyl-2-oleoyl-sn-glycero-3-phosphocholine) bilayer membrane model and solvated using the SPC (single point charge) water model. Na^+^/Cl^−^ ions for Na_v_1.5 and K^+^/Cl^−^ ions for K_v_1.5 were added to neutralize the systems, and then, NaCl or KCl was added to reach a concentration of 0.15 M in each case. K^+^ ions were placed at sites S2 and S4 of the selectivity filter (SF) and water molecules at sites S1 and S3 in K_v_1.5. No ions were located in the Na_v_1.5 SF. Systems were equilibrated by 20 ns in the NPT ensemble. Positional restraints of 1.0 kcal × mol^−1^ × Å^−2^ were applied to all protein and ligand atoms in the K_v_1.5 and Na_v_1.5 systems. At the same time, the same positional restraints were applied to the ions and water molecules placed in the SF of the K_v_1.5 channel. Temperature and pressure were kept constant at 300 K and 1.01325 bar, respectively, by coupling to a Nose-Hoover Chain thermostat [38] and Martyna-Tobias-Klein barostat [39]. The force field equation was integrated each at 2 fs in the MD simulations. Subsequently, positional restraints were removed, and 100 ns MDs were performed per system using a NPγT (semi-isotropic ensemble) with the constant surface tension of 0.0 bar Å. Hence, there are 300 ns of MD production for apo and holo systems from three replicas for each channel. To check MDs’ stabilization, root-mean-square deviation (RMSD) values of the proteins atoms were computed using TCL scripting in VMD v1.9.4a38 [40]. In total, 1.2 µs of the MD simulation were performed and analyzed. Structural and pore shaping changes were computed by root-mean-square-fluctuation (RMSF) of all atom residues in TCL scripting in VMD v1.9.4a38 [40] and HOLE software [41], respectively.

### 2.3. Flecainide Binding Site Characterization

For BS characterization purposes (Figure 1, step 1), 1000 frames of each MD were retrieved, and the residues within 5 Å of flecainide were presumed to belong to the BS, a definition that will be kept in all the manuscript. For each frame, Fpocket [42] physicochemical features (area, volume, hydrophobicity proportion, Monera hydrophobicity score [43] and proportion of nonpolar atoms) were computed in the BS, calculating the mean and standard deviation in each point of the three replicas per system. Moreover, in the BS, contacting residues were counted using TCL scripting in VMD v1.9.4a38 [40]. The flecainide interaction profile was obtained using PLIP [44,45].

### 2.4. Flecainide Binding Site Comparison

All MDs of the holo systems were concatenated and underwent a clustering analysis (Figure 1, step 2). This clustering was performed based on Fpocket physicochemical features computed in the previous section to retrieve representative structures for further analysis. The K-means algorithm with a euclidean distance implemented in R v4.1.2 and the NbClust package [46] was used to perform clustering and establish the optimum number of clusters, respectively. Two clusters were obtained per system (holo K_v_1.5 and holo Na_v_1.5), and the frame corresponding to the structure nearest to the computed K-means centroid was defined as representative structure of the cluster. Then, the representative structures were retrieved to compare their similarities by PocketMatch [47] using the previously defined BS (Figure 1, Step 2). The comparison between the representative structures of the four clusters of K_v_1.5 and Na_v_1.5 resulted in one pair of centroids with the best score. Then, PocketAlign software [48] was used to find amino acid correspondence in the pair of centroids with the best PocketMatch score.

### 2.5. Statistical Tests

The normality assumption was not satisfied by our data. For that reason, nonparametric statistical analysis was performed using R v4.1.2.

## 3. Results

This study compared the BS for flecainide in K_v_1.5 and Na_v_1.5 channels. In both channels, the PD provides BSs for most AADs and local anesthetics including flecainide BS [20,41,49,50,51,52,53,54]. Structurally, PD is composed of the helical segments S5, S6 and the loop connecting them (called P-loop). The latter contains the SF. In K_v_1.5, the SF sequence is TVGYG and provides a row of K^+^ coordination sites (called S1 to S4, from the extracellular to intracellular side of the cell membrane) [55]. In Na_v_1.5, the SF is asymmetric and composed of a ring of four residues DEKA (from Asp in DI to Ala in DIV) [20].

As a first step in our computational pipeline, tertiary structures of flecainide in the complex with K_v_1.5 and Na_v_.15 are needed. For K_v_1.5, a homology model reported by Marzian et al. [37] was used. In addition, we obtained the K_v_1.5-flecainide complex using an ensemble docking pipeline. We considered that flecainide exhibits a preferential action for K_v_1.5 in its open state with a Hill coefficient of about 1 [56]. Then, a unique ligand-BS or multiple non-cooperative ligand-BSs are anticipated. According to previous mutagenesis studies, residues from S6 helices (Ile-502, Val-505 and Ile-508) and the SF base (Thr-479, near to S4 K^+^) are involved in the action of flecainide [21,28]. This evidence allowed us to focus solely on the central cavity (also known as the inner cavity) to explore a single putative BS for flecainide.

For Na_v_1.5, the cryo-EM structure of rNa_v_1.5 in the complex with flecainide (PDB code: 6UZ3) and its apo form (PDB code: 6UZ0) was used [20]. The flecainide-rNav1.5 complex is assumed in an intermediate inactivated state [20]. This fulfills our requirements since flecainide stabilizes the channel inactivation state [57]. Although we are especially interested in human Na_v_1.5 (hNa_v_1.5), the latter shares about 94% of global sequence identity with its homolog in the rat. Then, the results obtained here can be extended to the hNa_v_1.5 channel.

Multiple MDs (three replicas) for K_v_1.5 and Na_v_1.5 channels in their apo and holo systems were executed during 100 ns. RMSD values are lower for K_v_1.5, but all trajectories are stabilized after about 50 ns (Appendix A). We also check for structural changes upon flecainide binding between apo and holo systems in K_v_1.5 and Na_v_1.5 by computing RMSF. These results are presented in Appendix A and evidence a global structure similarity between the apo and holo forms in both K_v_1.5 (Appendix A) and Na_v_1.5 (Appendix A). Moreover, in the PD (where flecainide binds), a local structural similarity between the apo and holo forms in K_v_1.5 (Appendix A) and Na_v_1.5 (Appendix A) is shown. The pore dimensions were also computed and compared between apo and holo systems (Appendix A). No differences were detected in pore size upon flecainide binding in K_v_1.5 (Appendix A). In the case of Na_v_1.5, the pore was slightly enlarged (Appendix A).

### 3.1. Flecainide Binding Mode and Interactions in K_v_1.5

In our predictions, flecainide occupies the K_v_1.5 central cavity (Figure 1a). Its piperidine moiety faces the pore but its trifluoromethyl groups protrude into the interface between subunits B and C. Residues in contact with flecainide (distance ≤ 5 Å and frequency ≥ 70%, Figure 1b) recorded along with MD simulations and those having interactions with this drug (number of interactions ≥ 20) are indicated in Figure 2a,b, respectively. The list of contacting residues includes Met-478.A, Thr-479.A, Thr-480.A, Gly-504.A, Val-505.A, Ile-508.A, Ala-509.A, Leu-437.B, Met-478.B, Thr-479.B, Thr-480.B, Ala-501.B, Ile-502.B, Gly-504.B, Val-505.B, Ile-508.B, Ala-509.B, Val-512.B, Ala-501.C, Ile-502.C and Val-505.C (Figure 1b and Figure 2a).

The interactions involved are indicated in Figure 2b. Trifluoromethyl groups interact via the halogen-bond with residues Thr-479.A, Thr-479.B, Ala-501.C, Ile-502.C and Ile-508.B (Figure 1b and Figure 2b). Hydrophobic interactions occurred with residues Thr-480.A, Thr-480.B, Val-505.A, Val-505.B, Ile-508.A, Ile-508.B, Ala-509.A and Ala-509.B (Figure 1b and Figure 2b). Residues Thr-480.A, Thr-480.B and Thr-480.C interact with flecainide through water bridges. This residue is placed at the inner mouth of the SF. Residues Thr-479, Ile-502, Val-505 and Ile-508 were previously noticed as relevant for the flecainide effect in K_v_1.5 using mutagenesis studies [21,28].

### 3.2. Flecainide Binding Mode and Interactions in Na_v_1.5

Similar to flecainide in K_v_1.5, this drug is placed in the central cavity of Na_v_1.5 below the SF (Figure 3a). In agreement with the cryo-EM flecainide-Na_v_1.5 structure (PDB code: 6UZ3), our MDs’ analysis confirms a number of residues that remain in close contact with flecainide: Leu-898.DII, Cys-899.DII, Val-933.DII, Phe-937.DII, Phe-1420.DIII, Ile-1456.DIII, Ile-1457.DIII, Phe-1461.DIII, Ile-1464.DIII and Phe-1762.DIV (Figure 3b and Figure 4a). Other residues also found in close contact include Gln-372.DI, Val-406.DI, Asn-930.DII, Leu-934.DII, Thr-1419.DIII, Thr-1711.DIV, Ser-1712.DIV and Val-1765.DIV (Figure 4a).

For Na_v_1.5 (Figure 3b and Figure 4b), we noticed that hydrophobic interactions involve aromatic residues (e.g., Phe-937.DII, Phe-1420.DIII, Phe-1461.DIII and Phe-1762.DIV). Other residues also contribute to hydrophobic interactions, including Gln-372.DI, Val-933.DII, Leu-934.DII, Leu-1464.DIII and Val-1765.DIV. Trifluoromethyl moieties interact with residues Thr-371.DI, Leu-898.DII, Cys-899.DII, Asn-930.DII and Ile-1456.DIII. Water bridges connect the residues Thr-371.DI, Gln-372.DI, Cys-899.DII, Gly-900.DII, Asn-930.DII, Thr-1419.DIII, Phe-1420.DIII and Ser-1712.DIV with flecainide. In agreement with Jiang et al. [20], residues Leu-898.DII, Cys-899.DII, Val-933.DII, Phe-937.DII, Phe-1420.DIII, Ile-1456.DIII, Ile-1457.DIII, Phe-1461.DIII, Ile-1464.DIII and Phe-1762.DIV remain in close contact with flecainide along MDs (Figure 3b and Figure 4a).

Interestingly, we detected π-stacking interactions between the central phenyl group of flecainide and residues Phe-937.DII, Phe-1420.DIII and Phe-1461.DIII (Figure 3b and Figure 4b). Most interactions occurred with Phe-1420.DIII.

### 3.3. Comparing Flecainide Binding Site

To compare flecainide pockets in K_v_1.5 and Na_v_1.5, we quantified physicochemical features using Fpocket in holo systems. We computed the volume (Figure 5a), area (Figure 5b), hydrophobicity proportion (Figure 5c), Monera hydrophobicity score (Figure 5d) and proportion of apolar atoms (Figure 5e) for each BS along MD simulations. Figure 5 shows that the flecainide pocket in K_v_1.5 exhibits a higher volume and area. This assumption was ratified by the Wilconsox Rank Sum test (Figure 5). This is not surprising since voluminous phenylalanine aromatic residues shape the flecainide BS in Na_v_1.5. When comparing the hydrophobic nature of both BSs, we found that both pockets are highly hydrophobic, Na_v_1.5 being the one that presents the highest hydrophobicity according to the Wilconsox Rank Sum test in the measurements of hydrophobicity proportion, Monera hydrophobicity score [43] and proportion of apolar atoms (Figure 5c–e). Moreover, the residues accounting for hydrophobicity differ in their side-chain size and volume between K_v_1.5 and Na_v_1.5 in the flecainide BS (Figure 1b and Figure 3b).

For a deep comparative analysis of the flecainide BS in the holo systems for Na_v_1.5 and K_v_1.5, we reduced the MD data by applying a clustering approach based on described physicochemical properties (Figure 5), as suggested by De Paris research [58]. The NbClust package performs a pre-running of K-means clusters calculation with different number of clusters, starting from 1. Then, NbClust computes 26 different indices. Each index determines an optimal number of clusters from previous K-means calculation. Finally, NbClust outputs the optimum number of clusters which is the most frequent value among the indices. In our case, we found that two clusters are the best choice for K_v_1.5 and Na_v_1.5 holo systems. For that reason, two centroids (representative frames) were retrieved from the 300 ns of each holo system. Regarding the flecainide-K_v_1.5 complex, cluster 1 and cluster 2 consist of 1506 and 1494 structures from a total number of 3000 frames, respectively. The centroid for the first cluster corresponds to frame 979 from the first MD replica. The frame number 113 of the second MD replica was defined as the centroid of the second cluster. Both were renamed *K_v_1.5_c1* and *K_v_1.5_c2*, respectively. Likewise, on the flecainide-Na_v_1.5 system, we obtained two clusters from a total number of 3000 frames where cluster 1 includes 1307 frames, and cluster 2 has 1693 frames. Frame number 43 from the third MD and number 458 from the first MD replica were computed as centroids and renamed, such as *Na_v_1.5_c1* and *Na_v_1.5_c2*, respectively. The BS residues from the centroids are described in Table 1 and Table 2.

Centroids were compared using PocketMatch (Table 3). *K_v_1.5_c2* and *Na_v_1.5_c2* had the best PocketMatch similarity score between different channels (78.61%) in their flecainide BS. This similarity score is similar to the one between BS from centroids of the same channel: *Na_v_1.5_c1* and *Na_v_1.5_c2* (83.27%).

The *K_v_1.5_c2* and *Na_v_1.5_c2* centroids were used to explore the hypothesis of a common structural pattern in flecainide BS between K_v_1.5 and Na_v_1.5 channels. BSs from *K_v_1.5_c2* and *Na_v_1.5_c2* structures were aligned using PocketAlign. As a result, a pairwise list of equivalent residues was obtained and, then, filtered using as cutoff a contact frequency equal to or greater than 70%. The amino acids resulting from PocketAlign analysis were distinguished by their physicochemical nature following PocketMatch classification. Ten residue equivalence was retrieved using PocketAlign (Figure 6 and Table 4). Residue equivalence numbers from 1 to 6 exhibit a similar physicochemical nature. Four of them (1, 2, 3 and 4) correspond to aliphatic, non-polar and uncharged amino acids (Figure 6a,b and Table 4). The other two (5 and 6) correspond to aliphatic, polar amino acids with hydroxy or mercapto groups. Equivalences from 7 to 10 do not share similar physicochemical features. Note that residue equivalence from 1 to 8 is well-fitted in the structural alignment (Figure 6c).

The matching residues, represented with surfaces in Figure 7, occupy the central cavity in both channels, the B-C subunit interface in K_v_1.5 and fenestration DII-DIII in Na_v_1.5. Figure 7 shows the common structural pattern. Equivalent residues from 1 to 6 form two elements in the flecainide BS: (1) a hydrophobic patch (Figure 7, see residue surface in color gray) comprised of residues Met-478.A, Ile-508.A, Ala-501.B, Ile-502.B in K_v_1.5 (Figure 7a) and Leu-898.DII, Val-933.DII, Ile-1456.DIII, Ile-1457.DIII in Na_v_1.5 (Figure 7b) and (2) a polar region (Figure 7, see residue surface in color blue) comprising Thr-479.A and Thr-479.B in K_v_1.5 (*zoom* in Figure 7a) and Cys-899.DII and Thr-1419.DIII in Na_v_1.5 (Figure 7b).

## 4. Discussion

When comparing flecainide poses in Na_v_1.5 and K_v_1.5 channels, we detected a consensus binding mode where the drug fits the central cavity with extensions to the lateral sides of the channel (Figure 1a and Figure 3a). In particular, the flecainide piperidine ring faces the inner cavity at the base of the SF; the aromatic moiety fits in the hydrophobic environment in the low levels of the inner cavity; and the trifluoromethyl moieties protrude to lateral sides (fenestration DII-DIII in Na_v_1.5 and subunit interface B-C in K_v_1.5). This binding mode resembles an angular conformation already predicted for long and flexible ligands in K_v_1.5 [59].

The protonated piperidine moiety sits in the cation attractive region close to the base of the SF [60] in both channels (Figure 1a and Figure 3a). In previous reports, the cationic groups of charged ligands bound to voltage-gated sodium and potassium channels are also attracted to the SF and occupy sites for permanent sodium ions in Nav channels [60]. In Na_v_1.5, the charged ammonium group exerts a pivotal role in flecainide-associated inhibition [57]. The sodium channel blocked by flecainide and two derivatives (one neutral and the other fully charged at a physiological pH) disclose that the blockade results from the interaction of the cationic form [57].

The comparative analysis of flecainide BS in K_v_1.5 and Na_v_1.5 highlighted that the presence of aromatic residues is a key distinguished feature in Na_v_1.5 (Figure 1b and Figure 3b). Importantly, three aromatic residues—Phe-937.DII, Phe-1420.DIII and Phe-1461.DIII—in Na_v_1.5 have π-stacking interactions with the phenyl ring of flecainide, and Phe-1762.DIV exhibits hydrophobic interactions with the piperidine moiety (Figure 3b and Figure 4b). Phe-1760 in hNa_v_1.5 (homolog to Phe-1762 in rNa_v_1.5) plays a role in local anesthetics and antiarrhythmic action [49,50]. In our predictions, the residue Phe-1762.DIV is close to the positively charged piperidine ring but establishes only hydrophobic interactions (Figure 4b). This residue usually establishes a cation-π interaction with charged drugs with some exceptions, including flecainide and ranolazine [61,62]. Our results are in agreement with experimental evidence probing that flecainide does not require the cation-π interaction with Phe-1760 for its binding and use-dependent blockade [62].

Previously, the influence of aromatic residues in the K_v_1.5 inner cavity was addressed by mutagenesis studies [21]. The substitutions of I502F and I508F in hK_v_1.5 increase the IC_50_ (164.49 µM ± 31.36 and 74.71 µM ± 5.37, respectively) compared to the wild-type (IC_50_ = 38.14 µM ± 1.06) [21]. Then, aromatic moieties disturb drug interactions in such positions. However, a similar mutation at position 505 (V505F) increases flecainide affinity for hK_v_1.5; the IC_50_ value decreases to 4.27 µM in HEK 293 cells [21]. A possible explanation was given by Eldstrom et al., 2007 [21]. They presume that substitution V505F could favor cation-π interactions with piperidine from flecainide. However, using our model, we suggest that the substitution of V505F could favor π-stacking interactions with the phenyl aromatic ring of flecainide. As shown in Figure 1b, the residue Val-505.C is placed in front of the phenyl ring of flecainide. Then, the localization of an aromatic residue close to the flecainide phenyl ring is likely to account for a higher affinity binding.

Considering all the previous discussion, we proposed that aromatic moieties (Phe-937.DII, Phe-1420.DIII and Phe-1461.DIII and Phe-1762.DIV) in Na_v_1.5 could explain flecainide’s higher affinity for this channel than K_v_1.5. Phe-1762.DIV is located in front of the flecainide piperidine ring but only establishes hydrophobic interactions (Figure 3b and Figure 4b). Residue Phe-1420.DIII and, to a minor extent, Phe-1461.DIII and Phe-937.DII are all placed near the flecainide phenyl ring (Figure 3b) [20]. Phe-1762.DIV has been extensively studied and recognized as a relevant interacting residue for local anesthetics and antiarrhythmics binding [49,50]. However, to the best of our knowledge the possible contribution of Phe-937.DII, Phe-1420.DIII and Phe-1461.DIII for high-affinity ligand binding in Na_v_1.5 is still not reported.

Our hypothesis for high-affinity flecainide binding in Na_v_1.5 is consistent with previous studies in potassium channel hERG, another high-affinity target of flecainide [25]. Melgari et al. revealed the importance of the aromatic residue Phe-656 as a principal binding determinant for flecainide. These authors found that mutant F656A in hERG increased he tIC_50_ 142-fold compared to the wild-type [25]. They argued that Phe-656 (from two different chains) interacts with two different moieties in the flecainide molecule. In detail, Phe-656 interacts with the benzamide moiety and the piperidine ring of flecainide [25]. We speculate that Phe-656 in hERG plays a similar role to Phe-1762.DIV in Na_v_1.5. In hERG and Na_v_1.5, respective aromatic residues Phe-656 and Phe-1762.DI are placed in front of the piperidine ring, respectively [20,21]. Interestingly, Phe-656 does not have a homolog aromatic residue counterpart in K_v_1.5, since, according to sequence alignments, Phe-656 corresponds to Val-512 in K_v_1.5 [63]. Our findings and those published by Melgari et al. [25] support the need for aromatic residues for high-affinity binding of flecainide in their preferred targets, Na_v_1.5 and hERG. Accordingly, we propose that the absence of aromatic residues in K_v_1.5 could explain the lower affinity for flecainide.

Although there was a distinctive presence of aromatic residues in Na_v_1.5, we found the flecainide BS shares similarities in both channels (Table 3). In detail, we identified similar geometrical and physiochemical properties. Both flecainide pockets are hydrophobic, although K_v_1.5 to a lesser extent (Figure 5c,e). This is a typical feature of multi-target drug BSs [64]. In agreement with previous observation, we revealed that most equivalent residues in flecainide BSs have a hydrophobic nature (Table 4 and Figure 6a,b).

As reported in Ehrt et al., the geometrical feature seems to be the most relevant determinant for promiscuous BS, increasing the chance of MTDL behavior [64]. As already mentioned, our comparative analysis of flecainide BSs revealed a similar geometry. The latter is denoted by the structurally equivalent residue pairs listed in Table 4 and shown in Figure 6. For most of them, the distance between their center of mass is lower than 2.5 Å (Figure 6c). The matching residues, represented with surfaces in Figure 7, occupy the central cavity in both channels, the B-C subunit interface in K_v_1.5 and fenestration DII-DIII in Na_v_1.5.

At the end of our computational workflow, we were able to identify a common structural pattern at flecainide BSs in Na_v_1.5 and K_v_1.5 channels (Figure 1). Equivalent residues from 1 to 6 form two common elements in the flecainide BS: (1) a hydrophobic patch (see residue surface in color gray, Figure 7) comprised by residues Met-478.A, Ile-508.A, Ala-501.B, Ile-502.B in K_v_1.5 (Figure 7a) and Leu-898.DII, Val-933.DII, Ile-1456.DIII, Ile-1457.DIII in Na_v_1.5 (Figure 7b), and (2) a polar region (Figure 7, see residue surface in color blue) comprising Thr-479.A and Thr-479.B in K_v_1.5 (Figure 7a) and Cys-899.DII and Thr-1419.DIII in Na_v_1.5 (Figure 7b). These two regions could be hot spots for a drug-protein interaction in atrial-selective MTDL design strategies for AF.

The residue Ile-508.A (included in the groups of aliphatic, non-polar or uncharged amino acids) (Table 4) exhibits a hydrophobic interaction along MD simulation (Figure 2b). Likewise, Val-933.DII (equivalent to Ile-508.A) in Na_v_1.5 presents a hydrophobic interaction with flecainide (Figure 4b). Regarding amino acid pairs that contain aliphatic, polar amino acids or a hydroxy or mercapto group, residue Thr-479.A exhibits halogen bonds’ interaction in K_v_1.5 (Figure 2b). Similar, its corresponding residue Cys-899.DII in Na_v_1.5 displays halogen bonds (Figure 4b). Four pairs of equivalent residues differ in their physicochemical nature according to PocketMatch classification (Table 4, numbers 7 to 10), but three of them establish similar interactions with flecainide. The K_v_1.5 residue Thr-480.B presents the hydrophobic interaction as well as its pair Phe-1420.DIII in Na_v_1.5. Val-505.B and Phe-1461.DIII display both hydrophobic interactions. Ala-509.B from K_v_1.5 and Phe-937.DII in Na_v_1.5 also present hydrophobic interactions with flecainide. Among these equivalent residue pairs, all the Phe from Na_v_1.5 are highlighted as part of the flecainide BS in the cryo-EM holo structure (PDB code: 6UZ3) [20]. These Phe have distinctive π-stacking interactions from K_v_1.5. Mainly, Phe-1420.DIII has the greatest π-stacking interaction. Some of the equivalent residue pairs, Thr-479, Ile-502, Val-505 and Ile-508, have recognized roles in flecainide binding in K_v_1.5 as determined by mutagenesis [21].

As discussed above, the flecainide interaction profile (with the equivalent residues) differs in K_v_1.5 and Na_v_1.5 by about 50% (only five from ten residues exhibit similar interactions). However, this usually occurs for promiscuous drugs where the protein-ligand interaction profiles are not well related [64]. We found interactions that have relatively frequent (π stacking) or rare (halogen bond) prevalence in the PDB database [65] in K_v_1.5 and Na_v_1.5 flecainide BSs. Water bridges could contribute to flecainide binding because of their relevant role in ligand affinity and selectivity [66].

## 5. Conclusions

Besides flecainide, most ADDs are promiscuous in their action mechanism, and little is known about the structural basis of such behavior. One possible explanation is that they can modify membrane properties [67]. However, the existence of ion channels’ structure in the complex with ligands in combination with mutagenesis studies reporting the binding site of antiarrhythmics suggests that direct binding is also playing a role in promiscuity activity, and it is likely that common structural similarities are present at the antiarrhythmic BSs.

In the context of AF, mathematical models revealed that a promiscuity drug action focused on atrial-selective targets could be a safer and more efficacious alternative approach to treat this disease [14,15]. We focused on atrial-selective targets K_v_1.5 and Na_v_1.5 and their common blocker flecainide; we performed a comparative study of BSs for this drug coupling docking, MD and pocket comparison.

Our study identified that K_v_1.5 and Na_v_1.5 shared several common residues required for flecainide binding. The majority of these counterpart residues have similar geometrical and physiochemical properties. This led us to propose a common structural pattern for flecainide BS. Such a common pattern consists of two matching areas: a hydrophobic patch and a polar region. We also found a distinctive feature only present in Na_v_1.5, which could be responsible for flecainide’s higher affinity in Na_v_1.5 versus K_v_1.5, which is the presence of aromatic residues and their associated putative π-stacking interactions. Another critical aromatic residue in Na_v_1.5 for local anesthetic and AAD binding is Phe-1762 (Phe-1760 in humans). This residue does not exhibit an aromatic counterpart in K_v_1.5. We speculate that Phe-1762 could also account for flecainide’s high-affinity binding in Na_v_1.5.

Our results are intended to be used in rational MTDL design and new discovery protocols. We propose that ligands in close contact with residues of the promiscuous BS found in Na_V_1.5 and K_v_1.5 channels would simultaneously exert a biological action in both channels. The protocol described here for the BS comparison could be extended to other systems, gaining knowledge of the structural basis of polypharmacological drugs’ effects.

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
