# Peer review of "Common Structural Pattern for Flecainide Binding in Atrial-Selective Kv1.5 and Nav1.5 Channels: A Computational Approach"

_pharmaceutics, 2022, doi:10.3390/pharmaceutics14071356_

Round 1

Reviewer 1 Report

Current drugs against Atrial fibrillation (AF) suffer from mutliple side effects, thus there is a need for developing atria-selective pharmaceutical tools. In the manuscript entitled “Common Structural Pattern for Flecainide Binding in Atri-2 al-selective Kv1.5 and Nav1.5 Channels: a Computational Ap-3 proach”, to gain better understanding of the binding sites (BS) of flecainide, a drug for AF treatments, with its target channels, Nav1.5 and Kv1.5, Mazola et al developed a computational workflow, identified a common structural pattern in flecainide BS for these two channels. The two flecainide pockets both contain a hydrophobic patch and a polar region, share a similar geometry. These pockets contain crucial residues known for flecainide-binding, as well as newly identified by this research. The newly identified crucial residues and protocols described here could find their applications in rational design of new pharmaceutical compounds for treatments of AF, even other diseases.

The manuscript is carefully written, the results are convincing. It is a pity that no wet-lab results were provided to confirm those crucial residues identified in this research.  

 I would suggest those newly identified residues be highlighted in Fig 6 and Fig 7.

Author Response

In agreement with Reviewer#1, the residues T1419.DIII and S1712.DIV in Nav1.5 and, the residues M478.A, A501.B, T480.B and A509.B in Kv1.5 were underlined in Fig. 6 and Fig.7. Please, find enclosed changes in Fig. 6 and Fig.7 in the re-submitted version.     

Reviewer 2 Report

In manuscript entitled “Common structural pattern for flecainide binding in atrial-selective Kv1.5 and Nav1.5 channels: a computational approach“ authors investigate a computational workflow for flecainide BS comparison in a flecainide-Kv1.5 docking model and a solved structure of flecainide-Nav1.5 complex. The workflow includes docking, molecular dynamics, BS characterization and pattern matching. The authors identified a common structural pattern in flecainide BS for these channels. The manuscript is very interesting according to this topic but there is still some question which is should be considered.

Minor revision:

1.       The list of authors is not complete, namely the authors stated AND in the end.

2.     Moderate English changes required.

Author Response

In agreement with observations of Reviewer#2, we checked the list of authors and improved our English writing.

Reviewer 3 Report

The authors have presented an interesting MD simulation platform that can be utilized for ligand-ion channel interactions, especially in base of common ligands or similar binding sites. Minor checking of English is required, for example: 

1.       Line 80: change ‘binding’ to ‘bind’.

2.       Line 90: replace ‘is choose’ with ‘was chosen’

3.       Line 223: replace ‘the pore suffers a slightly enlarged’ with ‘the pore was slightly enlarged’

TThe authors can strengthen their predictions by Proline/Alanine substitution of key residues at the binding site of felecainide. Is the interaction totally lost in such case? 

Author Response

Please, check the enclosed document

Reviewer 4 Report

The manuscript by Mazola et al. describes a computational multi-target drug design approach based on the structure and interaction of flecainide with both NaV1.5 and KV1.5 channels. The drug design approach is based on previous reports suggesting that a combined inhibition of two atrial channels NaV1.5 and KV1.5 produces a synergistic antiarrhythmic effect without ventricular effects. The manuscript is well-written, and the findings are interesting. However, experimental demonstration using mutational approaches and how that affects drug binding and channel currents would be required as the proof of concept.

Minor: Name of No. 8-affiliated author is missing in the manuscript.

Author Response

Please, check the enclosed document
